# An Economic Evaluation Framework for Government-Funded Home Adaptation Schemes: A Quantitative Approach

**DOI:** 10.3390/healthcare8030345

**Published:** 2020-09-18

**Authors:** Bo Zhang, Peng Zhou

**Affiliations:** 1School of Economics and Management, Beijing University of Chemical Technology, Beijing 100029, China; zhangbo2001_ren@hotmail.com; 2Cardiff Business School, Cardiff University, Cardiff CF10 3EU, UK

**Keywords:** disability, home adaptation, policy evaluation, independent living

## Abstract

The ability to live independently plays a crucial role in the mental and psychological wellbeing of the disabled. To achieve this goal, most governments spend a substantial budget on home adaptation projects. It has been observed that schemes with different target clients (residents versus landlords) are different in efficiency and effectiveness. To understand why and how these schemes differ in performance, this paper develops and applies a generic economic evaluation framework for government-funded home adaptation schemes. Based on the individual-level surveys collected in the United Kingdom, an empirical model was formulated to quantify the determinants for various performance indicators, including money costs, time costs and client satisfaction. Robust estimation procedures were applied to deal with the heteroscedasticity and outlier problems in the data. Results showed that a specialized independent living scheme dedicated to disability adaptations (e.g., the Physical Adaptations Grant, PAG) had higher efficiency and effectiveness than general-purpose schemes (e.g., the Disabled Facilities Grant, DFG), because the funds were provided to the landlords who had a stronger motivation to minimize the time cost in the short run and maximize the future rent potential in the long run. A “unified system” approach to adaptations should be a guiding principle for policy development, regardless of who actually delivers the service.

## 1. Introduction

In the United Kingdom, about 18 per cent of the population (11.6 million) have limited mobility because of health problems or disability (ONS, 2011) [1], lower than the United States figure (19.4%) but higher than the world average (15%) (WHO, 2018) [2]. The United Kingdom also has an ageing population with nearly 12 million people aged 65 and above (ONS, 2018) [3]. It is found that many incidents of the old, such as falls (Kendrick et al., 2014) [4] and stroke (Guimarães et al., 2020; Xu et al., 2018) [5,6], occur due to a mismatch between personal abilities and the home environment, resulting from the activities of living daily with a disability (Wahl et al., 2009) [7]. It has been a significant policy intention for the last two decades to deliver nursing and healthcare services at home via housing adaptations (Carlsson et al., 2017; Ewart and Harty, 2015; Kutty, 2010) [8,9,10], given the substantial health benefits and efficiency improvements of remaining at home (Evans, 2019; Lee and Edmonston, 2019; Petersson et al., 2008; Uppal, 2007; Pynoos and Nishita, 2003; Lilja et al., 2003) [11,12,13,14,15,16].

Most papers on home adaptation focus on the benefits to end users (Stark et al., 2017) [17], while our paper studies the issue from the policymaker/funder’s perspective. Therefore, we also discuss the costs and inputs of home adaptation projects. To do that, this paper theoretically developed a generic framework of economic evaluation of government-funded schemes. We then applied this theoretical framework to empirically evaluate the government-funded schemes which target the enabling of old/disabled people to live independently at home, based on 3670 surveys collected for funded home adaptation schemes across Wales. Our findings shed light on how public resources can be spent more efficiently in economic principle and more effectively in the caring sense.

According to Mackintosh and Heywood (2015) [18], almost half of all housing association households in England had a disabled member in need of home adaptations or modifications in 2014. Over £40 million has been spent annually by the subnational devolved government in Wales on home adaptation schemes to improve the living standards of disabled people. Among others, the Disabled Facilities Grant (DFG) is the most comprehensive scheme with the widest coverage and largest amount (Zhou et al., 2019) [19]. In addition, a specialized Physical Adaptations Grant (PAG) is funded through a standalone government budget dedicated to social housing (the Social Housing Grant). PAGs target the landlords rather than the tenants, on the basis that landlords have a longer scope and a stronger motivation to perform home adaptations to attract more tenants. The Independent Living Grant (ILG) is another alternative to the DFG, eligible for all stakeholders to apply but covers only 300 people per year. Around half of the applicants were waiting for DFGs at the time of grant. Rapid Response Adaptation (RRA) is designed for people over 60 years of age: RRAs are for people who are in hospital awaiting discharge, or have recently been discharged from hospital or are at risk of being admitted to hospital or into a care home.

Figure 1 compares the four government-funded home adaptation schemes in the United Kingdom (the boxes in the middle) in terms of who provides the funds (left side) and who applies for the funds (right side). The funding body can be governments at different administrative levels (e.g., subnational government, local authority) or non-government charitable bodies (e.g., local care and repair agency), while the applicants can be council tenants, private tenants, or registered social landlords.

Given the great input and wide impact of these funding schemes to the health of the disabled, this paper attempts to answer two practical questions:(i)How can we better measure the efficiency of these funding schemes?(ii)How can we improve the efficiency of these funding schemes?

## 2. Materials and Methods

This study develops a theoretical framework of economic evaluation, which is then empirically applied to a cross-sectional survey. 

### 2.1. Design

Despite a diverse portfolio of funding schemes, the Adaptations Review [20] in 2005 (also known as the Jones Review), the White Paper [21] and National Assembly for Wales [22,23] all identified a need to tackle delays, secure improvement in the provision of adaptations and reconsider adaptation performance indicators. To answer the research questions, we designed a generic economic evaluation framework and quantitative analysis toolbox to review the government-funded home adaptation schemes. 

### 2.2. Methods

There are two types of methodologies adopted in the existing literature on evaluating home adaptation schemes. The first type is theory driven. For example, Mackintosh and Heywood (2015) adopted theories of agenda setting and models of power and discourse to explore the structural and political factors in housing policy contributing to inadequate resource allocation for funding adaptation work in England [18]. Their study was embedded in an earlier strand of literature in the United States [24,25,26,27]. The other type of methodology is data-driven, following the philosophy of “let data speak”. Among others, Heywood (2004) showed the link between home adaptations and health state based on a large sample in England and Wales [28]. Heywood (2005) further provided a reflection on the “meaning of home” in evaluating the effectiveness of home adaptation, based on more in-depth interviews [29]. Recently, Ewart and Harty (2015) identified various biases of the adaptation provision based on English Housing Surveys over the last decade [9]. In fact, there is no such thing as purely theoretical or purely empirical methodology, since all studies resort to data to some extent. The fundamental difference is whether the purpose of data analysis is confirmatory or exploratory. In the theory-driven methodology, different datasets are used to verify the related theory, while the data-driven methodology usually involves collecting ad hoc datasets and drawing the conclusions after data mining [18].

This paper lies in the middle of the methodological spectrum, though it takes a more data-driven stance. There are two novelties of this proposed eclectic method: (i) the data analysis is systematically consistent to the conceptual framework derived from health economic theory; (ii) the data analysis is statistically strict by using robust econometric techniques.

In particular, the existing data-driven literature mainly used descriptive techniques such as cross-tabulation to relate the performance indicators to various socio-economic factors, such as ethnic groups, dwelling type, tenure, household type, disability type and income group. By contrast, the data analysis in this paper is more robust and systematic. All these factors are controlled at the same time when trying to quantify the effects of a particular determinant. The estimation procedure is also robust enough to deal with various data problems such as outliers and heteroscedasticity.

We propose a 4E framework commonly adopted in economic evaluation of health-related projects (Morris et al., 2012) [30]:**Efficiency**: if the inputs are minimized;**Effectiveness**: if the outcomes are maximized;**Equality**: if recipients have equal opportunities;**Ethics**: if ethical issues are handled properly.

Apart from Ethics, the other three Es can be evaluated using a quantitative analysis approach. This article uses econometric modelling techniques to address three aims: (i) aim 1: performance indicator (measurement); (ii) aim 2: performance evaluation (backward looking); (iii) aim 3: performance improvement (forward looking).

In econometric terminology, aim 1 defines the dependent variable of the model, aim 2 identifies the factors or regressors which affect the performance indicator(s), and aim 3 draws policy implications based on the results of aim 2. In particular, aim 1 is linked with quantifying the efficiency and effectiveness of the schemes, which we respectively define as scheme inputs (such as money and time) and outcome (such as client satisfaction), and they can then be explained by a set of factors in a regression model.

This policy evaluation framework was applied to evaluating the home adaptation schemes in the United Kingdom. The empirical analysis was based on the survey containing 4764 individual-level observations across 9 local authorities, carried out by various independent living adaptation (ILA) scheme providers on delivering the home modifications. The response rate was 75.06%. Only the two most popular ILA schemes (DFGs and PAGs) were included. In the survey, information was collected on the type of adaptation work, the cost of adaptations, customer satisfaction, days taken to complete adaptations and the demographic details of clients. Based on the individual-level information available, we were able to conduct an economic evaluation to inform the policymakers of the performance of existing ILA projects.

### 2.3. Analysis

First, we chose a set of performance indicators for the adaptation projects. Performance can be measured from either the supplier’s perspective or the demander’s perspective. In light of economic analysis, any outcome is a combined result from both the supply side (the developers) and the demand side (the clients or users). On the supply side, the performance indicator means the input of the ILA, both in terms of money and time costs. It focuses on the efficiency aspect of the evaluation. On the demand side, it means the outcome of the ILA, i.e., the recipient’s satisfaction, as a measure of living standard improvement. The survey asked the recipients to rank the outcome of the adaptation projects on a scale of 1–10, with 1 being the least satisfactory and 10 the most satisfactory. This measure focused on the effectiveness aspect of the evaluation. Either way, an indicator suitable for quantitative analysis should have the following two features: **Quantifiability**—the indicators must be measurable using some quantitative metric.**Commensurability**—the indicators must be used across all ILA schemes.

There might be several performance indicators and we could also develop a comprehensive performance index using a weighted average of these indicators. However, due to the lack of data and the problem of choosing the weights, we concentrated on the input performance indicators, which are defined as either money costs in pounds or time costs in days. We had both measures in our data, but the time costs were inconsistently defined across locations (detailed in Appendix A). With the performance indicators identified, we can now explicitly state the econometric model used in the analysis. Let us denote one of the three performance indicators as $y_{i}$. The subscript i indicates the individual case i, where i∈[1,N] and N is the sample size. We can then evaluate the performance of the ILA programs using the following econometric model:(1)$$y_{i}=\alpha +\beta_{1}\times s_{i1}+\beta_{2}\times s_{i2}+\ldots +\gamma_{1}\times d_{i1}+\gamma_{2}\times d_{i2}+\epsilon_{i}$$

The two sets of regressors $s_{ij}$ and $d_{ik}$ are the identified determinants for the performance indicator $y_{i}$, and the error term $\epsilon_{i}$ captures everything else the model treats as random. Our model is similar to Dudzinski et al. (1998) and Boström et al. (2018), but the former only focuses on the supply side of home adaptation projects [31] and the latter only focuses on the demand side [32]. Our framework combines both.

In particular, $s_{ij}$ stands for the supply-side factors such as:which type of adaptation is provided (details described in Appendix A);which ILA scheme it belongs to (e.g., DFG, PAG);which delivery method is used;which structure is used;other relevant factors related to the provision.

In contrast, $d_{ik}$ stands for the demand-side factors, such as:aggregate characteristics including economic development, institutional environment and other region-specific factors;individual characteristics including age, gender, race, disability type, family, etc.

It is straightforward that the relevant β and γ can be interpreted as the partial effects or marginal effects of $s_{ij}$ and $d_{ik}$ on $y_{i}$. The estimated partial effects of individual characteristics on $y_{i}$ can be used to inform policymakers about which group benefits the most from the ILAs and hence facilitate the discussion of equality issues. In practice, we used the natural logarithms of time and money costs in the model so that the partial effects could be interpreted as elasticities, i.e., how much percent improvement of efficiency/effectiveness due to a change in these factors.

### 2.4. Ethics

The surveys were collected by the service providers after home adaptations were delivered. Private individual information was anonymous and protected following EU Data Protection Regulation (GDPR) and ethically approved by the NHS.

All participants were informed about the survey purpose, voluntary nature, confidentiality, risks and benefits, and compensation, and provided with contact details of the person to whom they were to address questions.

## 3. Results

This section reports the empirical results to address the three aims outlined in Section 2.3. To recall, aim 1 defines the performance indicators (the dependent variable of the model), aim 2 identifies the factors that affect the performance indicators, and aim 3 draws policy implications based on the results of aim 2.

Table 1 descriptively summarizes the key performance indicators of the two most popular funded schemes, DFG and PAG, in terms of both inputs (money and time) and the outcome (satisfaction indicator). The data on the other two schemes (ILG and RRA) were not available, because the coverage of these two ad hoc schemes was very small.

Estimated economic evaluation models of different schemes are reported in Table 2 (least squares regression) and Table 3 (quantile regression). Different estimation procedures were used to provide a robustness check. The reasons for this are discussed in the next section. The estimation results showed pure contributions of different types of home adaptation and regional heterogeneity to the input costs and outcome performances.

As shown in Table 3, by both money and time standards, the costs were significantly lower for “rails”, and higher for “extensions”, reflecting the intrinsic differences due to adaptation type. Rural locations, such as Pembrokeshire, tend to have a lower money cost, mainly due to a lower general price level. After controlling for the differences in adaptation type and location, adaptations work funded by PAG schemes cost 0.96% less money and a 1.2% shorter time than DFG schemes, indicating a higher efficiency.

## 4. Discussion

Evidence shows that home adaptation schemes contribute to improving quality of life for the disabled and ageing populations (Lau et al., 2018; Hwang et al., 2011) [33,34], but most empirical studies are from the end-users’ rather than from the provider’s perspective (Tural et al., 2020) [35]. There are relatively fewer studies on the costs of home adaptations (Soares et al., 2020) [36]. In health economics, effectiveness analysis is often utilized to comparatively evaluate the performance of alternative arrangements (Cookson et al., 2017; Chiatti and Iwarsson, 2014) [37,38]. Among others, Curtis and Beecham (2018) estimated the cost composition of home adaptation schemes in the United Kingdom and found that DFG schemes spent more than they should due to a high staffing component accounting for 76% of the total cost [39]. Zhou et al. (2019) explored the causes of time delays of housing adaptations and also found that DFG schemes were understaffed and the resources were not enough to process the applications [19]. We found similar results in our regression results for DFG in comparison to its alternative, PAG. It is not a coincidence that PAGs are more efficient than DFGs in terms of both money costs and time costs—note that PAGs are provided to registered social landlords to adapt homes instead of to the tenants directly. Without government support, the landlords would not have much motivation to adapt homes for disabled people because they would have to bear the extra costs. Therefore, the landlords actually have a stronger motivation to seek funds for home adaptations in order to attract disabled tenants in the long run. In contrast, a short-tenure tenant with a disability may not want to bother with the adaptation as much as the landlord, because it may not seem worthwhile. Moreover, the landlords are also more motivated to choose the most efficient arrangements to minimize the short run-time costs of forgone rent. This improvement is possible because government funding schemes resolve the conflicts of interest between the landlords and the tenants regarding “who pays for it”. After the money issue is dealt with, the landlords are in a better position to decide what to do and how it should be done. This suggests that the government-funded schemes should be open to both end-users and property owners (landlords), and the latter would contribute to efficiency improvement due to a longer scope and more informed decision-making. It is arguably a better alternative to social care provided by specialized social care providers (Evans et al., 2019) [11].

In terms of outcome effectiveness (the last column of Table 3), the only significant factor is time cost, since the money cost is borne by the government. The estimated model suggests that a 10-day longer waiting time is associated with a 1.58% decrease in client’s satisfaction, other things being equal. Note that the evaluation based on satisfaction was very weak because over 75% of the responses reached maximal value, so it was not very sensitive. That was why we needed to use input indicators (money costs and time costs) to evaluate the projects to complement the lack of variations in output indicators. A preferred outcome indicator is Quality Adjusted Life Years (QALY) if expert assessments on health improvements are available (Soares et al., 2020; Bostrom et al., 2018) [32,36]. Combining the findings from the efficiency (money and time input) and effectiveness (output), the PAG scheme is believed to generate the highest client satisfaction with the least costs due to its appropriate applicant targeting.

There were some technical issues regarding the model estimation. Ordinary least squares (OLS) faced too many restrictive assumptions which might be violated due to the existence of heteroscedasticity and outliers. Diagnostic diagrams could provide intuitive eyeball tests for detecting heteroscedasticity (Figure 2) and outliers (Figure 3). In Figure 2, the variance of the regression residuals was smaller as the fitted value got larger, suggesting a relationship between the variance and some of the regressors, i.e., heteroscedasticity. In Figure 3, the average values of leverage and the (normalized) residuals squared, the points above the horizontal line had higher-than-average leverage; points to the right of the vertical line had larger-than-average residuals. We could easily spot the problematic observations in the upper right area, which had both high leverages and high residuals. This can also be seen by the high standard deviations of the performance indicators in Table 1. The influential observations (outliers) can lead to very unstable and unreliable estimates.

One easy way to deal with the heteroscedasticity problems is to use robust standard errors (Huber−White sandwich estimators). It keeps all the OLS point estimates of β and γ unchanged, but the standard errors are estimated in a more robust way. As seen in Table 2, some regressors became insignificant or less significant when robust standard errors were used. However, this procedure has its limitations in handling outliers, and there are more systematic and efficient approaches to deal with both problems at the same time, such as quantile regression. The estimation results based on least squares criterion (i.e., OLS estimates, OLS estimates with robust standard errors and robust regression estimates) are reported for money costs in Table 2, while the quantile regression results for all three performance indicators (money costs, time costs and satisfaction) are compared in Table 3.

## 5. Conclusions

This paper provides both a theoretical framework and a practical guide for the economic evaluation of government-funded home adaptation schemes for disabled and elderly populations. We applied this framework to review the performance of the prevailing home modification funding schemes in the United Kingdom. A set of performance indicators were defined, based on health economic theory and data availability, measuring the input efficiency (money costs and time costs) and the outcome effectiveness (satisfaction). A quantitative regression model was formulated to estimate the determinants for these performance indicators. To deal with the heteroscedasticity and outlier problem undermining the validity of OLS estimates, we used robust regression and quantile regression techniques to provide more reliable results.

According to the estimated regression models, it was shown that, after controlling for the differences in adaptation type and location, PAG outperformed DFG in terms of both money costs and time costs, while client satisfaction rises with a lower time cost. Compared to DFGs, PAGs gain higher efficiency and effectiveness due to a stronger motivation to minimize the costs from the registered social landlords.

If other individual characteristics, such as gender and race, were included in the model, then equality could also be evaluated. Equality issues were addressed by the qualitative research by Bibbings et al. (2015), who contended that waiting time and money cost depended very much on a person’s tenure and age as well as location, but not on race or gender [40]. The level of support for the idea of a unified system reflects the fact that stakeholders feel strongly that people ought to receive the same level of service no matter who they are. Therefore, it is arguable that a “one system” approach to adaptations should be a guiding principle for policy development, regardless of who actually delivers the service. Under this unified system, recipients of adaptation services could expect similar levels of service no matter what their circumstances may be. It could encourage greater consistency in terms of means testing, information provision and waiting times.

## Figures and Tables

**Figure 1 healthcare-08-00345-f001:**
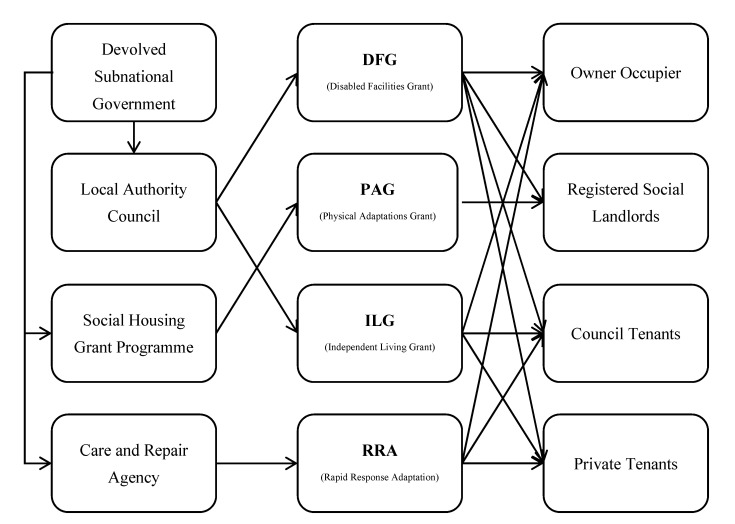
Comparison of prevailing independent living schemes in the United Kingdom. Note: the left blocks are the funders of the schemes in the middle blocks, and the right blocks are the targeted applicants.

**Figure 2 healthcare-08-00345-f002:**
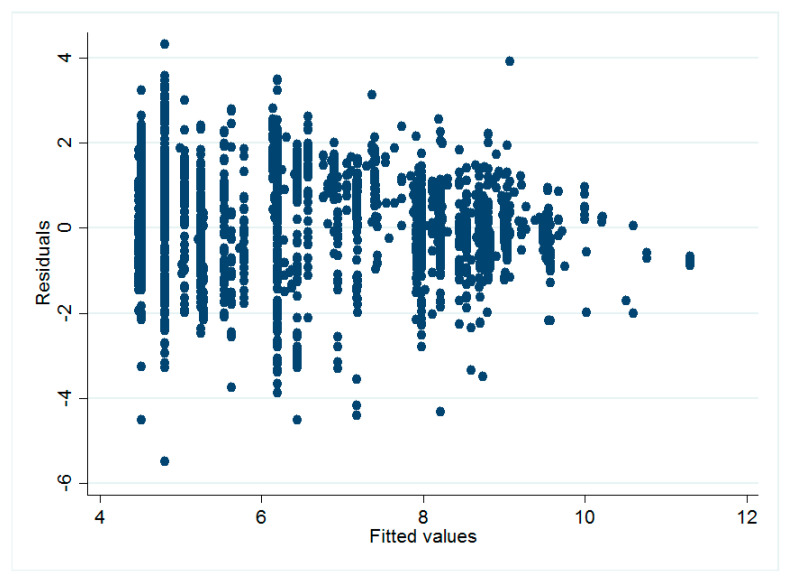
Relationships between residuals and fitted values (heteroscedasticity).

**Figure 3 healthcare-08-00345-f003:**
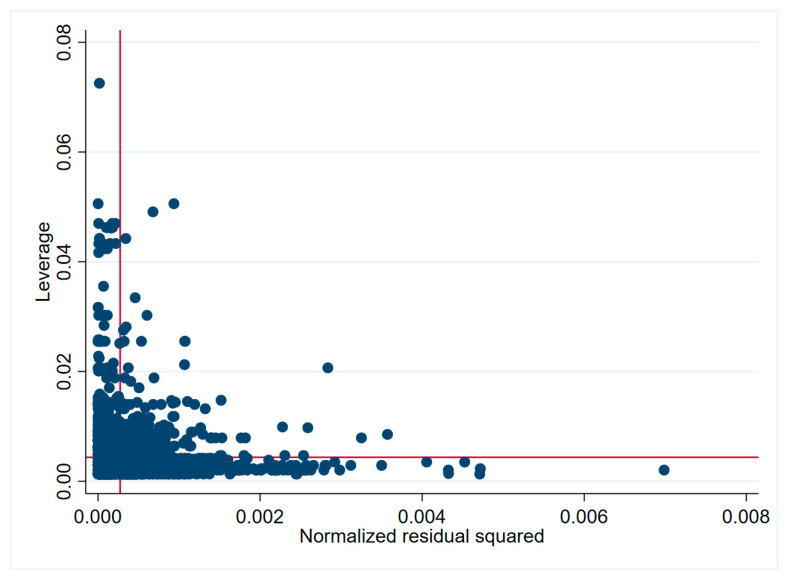
Relationships between residuals and leverages (outliers). Figure notes: leverage is a measure to identify the outlier observations. It ranges between 0 and 1, and a higher leverage score indicates a more influential observation.

**Table 1 healthcare-08-00345-t001:** Descriptive statistics of performance indicators.

Performance	Variable	Mean	Std. Dev.	25%	Median	75%
Money Input (in £)	Grant	3887	5712	209	2549	4514
Time Input (in days)	Total days	177	152	65	143	257
	contact−approval	87	96	15	54	132
	approval−finish	56	75	10	30	70
Outcome	Satisfaction	9.74	0.94	10	10	10

**Table 2 healthcare-08-00345-t002:** Determinants for efficiency measured by money costs (least squares).

Regressor	Log Money Cost (in £)		
	OLS	Huber−White	Robust Reg
access	0.560 ***	0.560 **	0.643 ***
shower	0.811 ***	0.811 **	0.838 ***
lift	0.718 ***	0.718	0.783 ***
rails	−1.131 ***	−1.131 *	−1.306 ***
stairlift	0.510 ***	0.510 *	0.471 ***
toilet	−0.018	−0.018	−0.004
extension	1.828 ***	1.828 ***	1.888 ***
hoist	0.524 **	0.524 ***	0.533 **
misc	−0.833 ***	−0.833	−1.144 ***
PAG	−1.734 ***	−1.734 ***	−1.480 ***
Cadwyn	−0.086	−0.086	0.086
Caerphilly	−0.706 ***	−0.706 ***	−0.676 ***
Glamorgan	0.117	0.117 *	0.095
Merthyr	0.000	0.000	0.000
Newport	−0.389 ***	−0.389 ***	−0.401 ***
Neath & PT	0.000	0.000	0.000
Pembroke	−2.471 ***	−2.471 ***	−2.307 ***
Torfaen	0.000	0.000	0.000
constant	8.111 ***	8.111 ***	8.069 ***
observations	3670	3670	3670
R^2^	0.658	0.658	0.705
R^2^ adjusted	0.656	0.656	0.703

Table notes: the reference location is Bridgend (V2C). * *p* < 0.05, ** *p* < 0.01, *** *p* < 0.001. The second column uses the Huber−White robust estimators of variance. The third column applies an iterative regression procedure to address the outlier issue based on Cook’s distance (“rreg” in stata).

**Table 3 healthcare-08-00345-t003:** Summary of the efficiency evaluation (quantile estimation).

Regressor	Input		Outcome
	Money Cost	Time Cost	Satisfaction
access	0.717 ***	0.202 ***	−0.244
shower	0.838 ***	0.052	−0.051
lift	0.775 ***	−0.135	0.000
rails	−1.545 ***	−0.346 ***	0.000
stairlift	0.441 ***	−0.005	−0.137
toilet	0.140 *	0.127	0.000
extension	2.066 ***	1.243 ***	−0.18
hoist	0.553 ***	−0.012	0.000
misc	−1.196 ***	0.141	0.032
PAGs	−0.952 ***	−1.203 ***	-
constant	7.801 ***	5.310 ***	11.034 ***
money cost	-	-	−0.044
time cost	-	-	−0.158 *

Table notes: * *p* < 0.05, *** *p* < 0.001. The location dummies are omitted. The regression of satisfaction is based on a smaller sample (*N* = 622) due to data availability.

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
