# Peer review of "An Economic Evaluation Framework for Government-Funded Home Adaptation Schemes: A Quantitative Approach"

_healthcare, 2020, doi:10.3390/healthcare8030345_

Round 1
Reviewer 1 Report
I have reviewed the paper
An Economic Evaluation Framework for Government-Funded Home Adaptation Schemes: A Quantitative Approach
Manuscript ID: healthcare-932831
I have previously reviewed this paper with the registration
Manuscript ID: healthcare-876207
My result was: rejected
I have verified that the paper was rejected and that the authors have resubmitted it with numerous improvements.
Even so, I continue to detect some very important weaknesses that must be improved.
Introduction
Authors should increase the number of references in the introduction.
This correction has already been raised previously, and is not resolved.
If the authors propose an analysis model, they must incorporate a large number of references. Some may be older, but many current should appear.
Figure 1 should improve it. The authors must put the lines with arrows. In the figure there are some arrows, but there are other lines without arrows.
line 69. An isolated epigraph should not appear
Results
Table 2 should not appear on two sheets.
Table 3 should not appear on two sheets.
Discussion
The discussion must be deeply improved.
This correction has already been raised previously and is unresolved.
The authors do not dispute their results with previous studies.
The discussion is a section that should have a large number of updated references.
The authors must redo this section.
References
Authors should add more references, about 20-30 more references for an article that could be published in a journal such as Healthcare
Authors must incorporate a significant number of references from the last 3 years.
Author Response
Responses to Referee 1
C1. Introduction. Authors should increase the number of references in the introduction. This correction has already been raised previously, and is not resolved. If the authors propose an analysis model, they must incorporate a large number of references. Some may be older, but many current should appear.
R1. I have now incorporated another 10 recent studies into the introduction. There is an extensive literature on home adaptation, including those published in Healthcare such as Carlsson et al (2017), Evans (2019) and Lee and Edmonston (2019). We have cited them in our paper. To substantiate the literature review in introduction, we also compare our paper with the existing publications. Most papers on home adaptation focus on the benefits to end users, while our paper studies the issue from the policymaker/funder’s point of view. Therefore, we also discuss the costs and inputs of home adaptation projects. We hope our paper is a good addition of this important topic to Healthcare.
- Carlsson, G.; Nilsson, M.; Ekstam, L.; Chiatti, C.; Fange, A. Falls and fear of falling among persons who receive housing adaptations—results from a quasi-experimental study in Sweden. Healthcare 2017, 5, 66; doi:10.3390/healthcare5040066.
- Evans, S., Waller, S., Bray, J., Atkinson, T. Making homes more dementia-friendly through the use of aids and adaptations, Healthcare 2019 7(1): 43.
- Lee, S.M.; Edmonston, B. Living Alone Among Older Adults in Canada and the U.S.. Healthcare 2019, 7, 68.
- Living longer, 2018. Retrieved from https://www.ons.gov.uk/peoplepopulationandcommunity/birthsdeathsandmarriages/ageing /articles/livinglongerhowourpopulationischangingandwhyitmatters/2018-08-13
- Petersson I, Lilja M, Hammel J, Kottorp A. Impact of home modification services on ability in everyday life for people ageing with disabilities. Journal of Rehabilitation Medication. 2008; 40(4): 253-260. doi:10.2340/16501977-0160
- Kendrick D, Kumar A, Carpenter H, Zijlstra GAR, Skelton DA, Cook JR, Stevens Z, Belcher CM, Haworth D, Gawler SJ, Gage H, Masud T, Bowling A, Pearl M, Morris RW, Iliffe S, Delbaere Exercise for reducing fear of falling in older people living in the community. Cochrane Database of Systematic Reviews 2014, Issue 11. Art. No.: CD009848. DOI: 10.1002/14651858.CD009848.pub2.
- Xu, Tianma; Lindy Clemson; Kate O'Loughlin; Natasha A. Lannin; Catherine Dean and Gerald Koh. 2018. "Risk Factors for Falls in Community Stroke Survivors: A Systematic Review and Meta-Analysis." Archives of Physical Medicine and Rehabilitation, 99(3), 563-73.e5.
- Wahl, H. W., Fange, A., Oswald, F., Gitlin, L.N., & Iwarsson, S. The home environment and disability-related outcomes in aging individuals: What is the empirical evidence? Gerontologist 2009, 49, 355–367.
- Guimarães, Moema; Ms Maiana Monteiro; Rafael Tito Matos; Ms Cláudia Furtado; Helena Fraga Maia; Lorena R. S. Almeida; Jamary Oliveira Filho and Elen Beatriz Pinto. 2020. External Validation of the Recurrent Falls Risk Scale in Community-Dwelling Stroke Individuals. Journal of Stroke and Cerebrovascular Diseases, 29(9), 104985.
- Stark S, Keglovits M, Arbesman M, Lieberman D. Effect of Home Modification Interventions on the Participation of Community-Dwelling Adults With Health Conditions: A Systematic Review. American Journal of Occupational Therapy. 2017, 71(2):7102290010p1-7102290010p11. doi:10.5014/ajot.2017.018887
C2. Figure 1 should improve it. The authors must put the lines with arrows. In the figure there are some arrows, but there are other lines without arrows.
R2. Done it.
C3. Line 69. An isolated epigraph should not appear.
R3. Done it.
C4. Result. Table 2 should not appear on two sheets. Table 3 should not appear on two sheets.
R4. Done it.
C5. Discussion. The discussion must be deeply improved. This correction has already been raised previously and is unresolved. The authors do not dispute their results with previous studies. The discussion is a section that should have a large number of updated references. The authors must redo this section.
R5. We have now rewritten Discussion section and added many recent publications in occupational therapy and home adaptations to compare with our own results. The new references include:
- Soares M, Sculpher M, Claxton, K. Health opportunity costs: Assessing the implications of uncertainty using elicitation methods with experts. Medical Decision Making, 2020, 40(4): 448-459.
- Cookson, Richard; Andrew J. Mirelman; Susan Griffin; Miqdad Asaria; Bryony Dawkins; Ole Frithjof Norheim; Stéphane Verguet and Anthony J. Culyer. Using Cost-Effectiveness Analysis to Address Health Equity Concerns. Value in Health 2017, 20(2), 206-12.
- Zhou, W., Oyegoke, A.S. & Sun, M. Service planning and delivery outcomes of home adaptations for ageing in the UK. J Hous and the Built Environ 34, 365–383 2019. https://doi.org/10.1007/s10901-017-9580-3
- Hwang, Eunju; Linda Cummings; Andrew Sixsmith and Judith Sixsmith. Impacts of Home Modifications on Aging-in-Place. Journal of Housing for the Elderly 2011, 25(3), 246-57.
- Lau G, Yu M, Brown T, Locke, C. Clients’ Perspectives of the Effectiveness of Home Modification Recommendations by Occupational Therapists. Occupational Therapy in Health Care 2018, 32(3): 230-250.
- Tural E, Lu D, Cole D. Factors predicting older Adults’ attitudes toward and intentions to use stair mobility assistive designs at home. Preventive Medicine Reports 2020, 18, 1010-1082.
- Curtis L, Beecham J. A survey of local authorities and home improvement agencies: identifying the hidden costs of providing a home adaptations service. British Journal of Occupational Therapy 2018, 81(11): 633-640.
- Chiatti C, Iwarsson S. Evaluation of housing adaptation interventions: integrating the economic perspective into occupational therapy practice, Scandinavian Journal of Occupational Therapy 2014, 21(5): 323-333, DOI: 10.3109/11038128.2014.900109
- Boström L, Chiatti C, Thordardottir B, Ekstam L, Fange A. Health-Related Quality of Life among People Applying for Housing Adaptations: Associated Factors. International Journal of Environmental Research and Public Health, 2018, 15(10): 21-30.
- Stark S, Keglovits M, Arbesman M, Lieberman D. Effect of Home Modification Interventions on the Participation of Community-Dwelling Adults With Health Conditions: A Systematic Review. American Journal of Occupational Therapy. 2017, 71(2):7102290010p1-7102290010p11. doi:10.5014/ajot.2017.018887
C6. References. Authors should add more references, about 20-30 more references for an article that could be published in a journal such as Healthcare. Authors must incorporate a significant number of references from the last 3 years.
R6. Following the referee’s comment, we have now added 10 more references from the recent decade in Introduction section. We also compare our findings in Discussion section with another 10 references from the recent literature. We have cited all articles from Healthcare which are related to home adaptations, but there are only a few. Therefore, we hope our paper can add another tribute to this important strand of literature in Healthcare.

Reviewer 2 Report
The text of the article has been improved, the description of methods is clearer, some issues are better explained. Anyway, there are still weaknesses in some points.
I still have an impression, that this article is much more focused on a methodology than on issues important for a health and health care. Of course, it is understandable that the methodology of research is very important area, also in health care, but the significance of this study results for the health care area is limited.
Line 19-23: „higher” suggests comparison – higher than what? Results do not indicate a reason of higher efficiency and effectiveness, so this sentence formulation is not proper.
In the lines 34-37 a description what was done can be found. But in the study, there is not needs of the disabled people evaluation as stated (there is not such formulation later, when research questions and aims are presented). The indication „how public resourses can be spent(..) more effectively in caring sense” is questionable as well, as in the descriptive statistics can be seen that probably in more than 75% cases the satisfaction indicator reaches maximal value (so it is not very sensitive).
Line 157: two schemes are included in the study, not four - isn’t truth?
Line 165: shouldn’t be γ’s as well?
Line 179: „the three” doubled
Lines 213-222: a similar remarks are in the Conclusion part (they fit better to the Conclusions or Discussion in fact)
Author Response
Responses to Referee 2
C1. The text of the article has been improved, the description of methods is clearer, some issues are better explained. Anyway, there are still weaknesses in some points. I still have an impression, that this article is much more focused on a methodology than on issues important for a health and health care. Of course, it is understandable that the methodology of research is very important area, also in health care, but the significance of this study results for the health care area is limited.
R1. Yes, as the referee pointed out, our study looks like a methodological toolbox for economic evaluations of home adaptation, but that is just an application of our proposed analytical framework of economic evaluation of any government-funded healthcare schemes. This is
As you can see from our literature review in the revised Introduction section, most papers on home adaptation focus on the benefits to end users, while our paper studies the issue from the policymaker/funder’s point of view. Therefore, we also discuss the costs and inputs of home adaptation projects. We hope our paper is a good addition of this topic to Healthcare.
C2. Line 19-23: „higher” suggests comparison – higher than what? Results do not indicate a reason of higher efficiency and effectiveness, so this sentence formulation is not proper.
R2. We have completed the statement now. As shown in Table 2 and Table 3, the estimated coefficients of the dummy variable for PAGs are negative and significant (-1.734*** and -0.952***), which suggest that the monetary and time costs are lower for any given adaptation project—or a higher efficiency and (cost) effectiveness.
C3. In the lines 34-37 a description what was done can be found. But in the study, there is not needs of the disabled people evaluation as stated (there is not such formulation later, when research questions and aims are presented).
R3. The Introduction section is now rewritten. The sentence in question is now corrected as “We then apply this theoretical framework to empirically evaluate the government-funded schemes targeting at the old/disabled people to live independently at home, …”
C4. The indication „how public resourses can be spent(..) more effectively in caring sense” is questionable as well, as in the descriptive statistics can be seen that probably in more than 75% cases the satisfaction indicator reaches maximal value (so it is not very sensitive).
R4. The referee is quite right that the satisfaction indicator is not very sensitive, and this is indeed what we found in regression results of Table 3 (the last column). The only significant factor that affects satisfaction is time cost (-0.158*): a 10-day longer waiting time is associated with a 1.58% decrease in client’s satisfaction. This is exactly why we need to use input indicators (money costs and time costs) to evaluate the projects to complement the lack of variations in output indicator (satisfaction). To appreciate the referee’s comment, we have emphasized this point in the last paragraph of Results section.
C5. Line 157: two schemes are included in the study, not four - isn’t truth?
R5. Corrected it. Thanks.
C6. Line 165: shouldn’t be γ’s as well?
R6. Corrected it. Thanks.
C7. Line 179: „the three” doubled
R7. Corrected it. Thanks.
C8. Lines 213-222: a similar remarks are in the Conclusion part (they fit better to the Conclusions or Discussion in fact)
R8. Thanks for pointing this out. After adjusting these lines, we distinguish between reporting the results in Results section and discussing the results in Discussion section. In addition, we have rewritten and added more discussions in connection with the recent literature to compare with our results.

Round 2
Reviewer 1 Report
I have reviewed the paper, second revision
An Economic Evaluation Framework for Government-Funded Home Adaptation Schemes: A Quantitative Approach
Manuscript ID: healthcare-932831
I have verified that the authors have incorporated my suggestions and corrections.
For my part, this paper is for publication when properly edited.
This manuscript is a resubmission of an earlier submission. The following is a list of the peer review reports and author responses from that submission.
Round 1
Reviewer 1 Report
Extremely well-written. Line 24 sentence begins with 18.2% -- might revise that sentence not to begin with a number. Line 35 "According to [9] -- give the authors and year instead then add [9] at end.
other than that, great paper!
Reviewer 2 Report
Comments in the attachment

Reviewer 3 Report
I have reviewed the paper An Economic Evaluation Framework for Government-Funded Home Adaptation Schemes: A Quantitative Approach
Manuscript ID: healthcare-876207
Abstract
The abstract does not meet the Journal's standards.
The abstract must be adapted to the norms of the Journal
https://www.mdpi.com/journal/healthcare/instructions
The abstract is not clear. The abstract is the first presentation of the paper. The reader must clearly identify the topic, the lack of research on the topic, the objective, the impact of the derived results, the methodology and the main results together with the conclusions. No sub-headings required. but it is necessary that the writing is well organized.
Introduction
The introduction is not properly stated. It has scientific errors.
For example, you should never start with a number.
18.2% of the population (11.6 million) in the United Kingdom have limited activities because of 24 health problem or disability
You should also not put a reference in a parenthesis, etc.
Line 26
but higher than the 25 world average (15%, [2]).
Rapid Response Adaptation (RRA) is not defined
Methods
The methodological description does not clear.
Results
The results section should introduce it, perhaps remembering the general objective
You can state the results in the order that you respond to the objectives.
Discussion
The discussion should improve it in depth.
You should introduce the overview of this job.
You should organize the findings and clearly discuss them with other studies.
References
The number of references are few. references are out of date.
The authors must make a deep review of the literature to give broad support to this paper.
